# Effect of the Support on Rhenium Carbide in the Hydrodeoxygenation of Guaiacol as Lignin-Derived Model Compound

Elodie Blanco [1,2,3,*], Ana Belén Dongil [4,*], Isaac Tyrone Ghampson [5] and Néstor Escalona [2,3,6,7]

1    Departamento de Ingeniería y Gestión de la Construcción, Pontificia Universidad Católica de Chile, Santiago 7820436, Chile
2    Departamento de Ingeniería Química y Bioprocesos, Pontificia Universidad Católica de Chile, Santiago 7820436, Chile
3    ANID (Agencia Nacional de Investigación y Desarrollo de Chile)—Millennium Science Initiative Program- Millennium Nuclei on Catalytic Process towards Sustainable Chemistry (CSC), Santiago 7820436, Chile
4    Instituto de Catálisis y Petroleoquímica, CSIC (Consejo Superior de Investigaciones Científicas), Cantoblanco, 28049 Madrid, Spain
5    Department of Applied Chemistry for Environment, Graduate School of Urban Environmental Sciences, Tokyo Metropolitan University, 1-1 Minami-Osawa, Hachioji, Tokyo 192-0397, Japan
6    Departamento de Química Física, Facultad de Química y de Farmacia, Pontificia Universidad Católica de Chile, Santiago 7820436, Chile
7    Centro de Investigación en Nanotecnología y Materiales CIEN-UC, Pontificia Universidad Católica de Chile, Santiago 7820436, Chile
*    Correspondence: elblanco@uc.cl (E.B.); a.dongil@csic.es (A.B.D.)

**Abstract:** The effect of support on the formation of rhenium carbide in the hydrodeoxygenation (HDO) of guaiacol as a lignin-derived compound was evaluated. Catalysts were prepared by incipient wetness impregnation, carburized at 650 °C under a mixture 25/75 of ethylene/hydrogen, and characterized by XRD, $N_2$-physisorption, TPR, TPD of $NH_3$ or isopropylamine (IPA), and XPS. The results have demonstrated that the support choice affects the carburization of rhenium. Indeed, over non-acidic support (e.g., $SiO_2$), limited carburization of the metal was observed, while over $ZrO_2$ and $Al_2O_3$, carburization of rhenium occurred, and different carbidic species were obtained depending on the surface and chemical identity of the support. The rhenium carbide species obtained over $ZrO_2$ was the most active and selective toward benzene (22% at 98% of conversion).

**Keywords:** HDO; rhenium carbides; acidity

## 1. Introduction

Increasing concerns about climate change and fossil fuel depletion have intensified the development of renewable alternatives to progressively replace current energy resources in the last few decades. Biomass is recognized as the only raw material that can replace both fuel and chemical production. Biomass is 100% carbon-neutral since it absorbs $CO_2$ for its growth. It can be transformed by pyrolysis and hydrogenolysis into platform molecules that can be further upgraded into chemical commodities with high added value such as plastics, resin, and chemicals for the pharmaceutical industry. Pyrolysis of biomass produces bio-oil, which is composed of several oxygenated-aromatic monomers [1] and possesses poor calorific capacity, high viscosity, and poor stability over time due to high functionalization [2]. Thus, the oxygen content must be decreased in order to valorize bio-oil. Hydrodeoxygenation (HDO) is a widely used process in that regard [3,4].

Hydrodeoxygenation studies typically involve the use of model compounds such as guaiacol and phenol. Many catalysts have been evaluated, with early studies focusing

on metal sulfides that favor the formation of aromatics such as benzene and toluene xylene (BTX) [5].However, metal sulfides require the presence of a sulfiding agent in the feed to prevent phase transformation, leading to interest in non-sulfide catalysts. In that regard, noble metal catalysts have generally been reported as the most effective with regards to producing fully hydrogenated compounds [6,7]. Non-noble metallic catalysts, principally Ni, have also been shown to produce high yields of fully hydrogenated compounds, although they are not as stable as noble metal catalysts [8–10]. Other active phases have been investigated, such as oxides [11–13], nitrides [14,15], phosphides [16–18], and carbides [19–21]. In particular, carbides have recently garnered a lot of attention, with $Mo_2C$ being shown to be highly active for several reactions [22–25]. Additionally, the catalytic behavior of $Mo_2C$ can be easily tuned by adjusting the oxygen/carbon content on the metal. In previous work on the HDO of guaiacol over different metal carbides, rhenium carbide (ReC) was shown to be more active than $Mo_2C$ and more selective toward BTX [26]. Subsequent studies led to the optimization of carburization conditions to prepare the carbides using activated carbon as a support [27,28].

Carbon-based materials are often chosen as supports for HDO reactions (especially when carbides are used as the active phase). Using metal oxide supports with different acidic properties, such as $ZrO_2$ or $Al_2O_3$, could expand the versatility of the carbides either through direct participation of the supports in the reaction or through modification of the electronic character of the carbidic phase. This is true for other active phases such as metal sulfides [29], metal oxides [13], and noble metals [30], among others. In those studies, it was shown that CoMoS supported on $ZrO_2$ led to an efficient conversion of guaiacol into deoxygenated hydrocarbons with totally different selectivities compared to CoMoS supported on $Al_2O_3$ [29]. A similar trend was observed in the HDO of cresol over a $MoO_3$ supported on $ZrO_2$, where the support stabilized partially reduced, coordinatively unsaturated (CU) sites, leading to high yields of toluene [13]. Finally, in the HDO of phenol over Pd catalysts, it was reported that the use of an oxophilic support such as $ZrO_2$ favored selectivity towards benzene [30]. It would be interesting to determine if those reported findings manifest when metal carbides are used as the active phase. Indeed, in the case of $Mo_2C$ supported on $Al_2O_3$ or $ZrO_2$ for a methanol steam reforming reaction, the methanation reaction was suppressed due to the ability of $ZrO_2$ to retain methoxy groups while resisting the formation of hydroxyl groups on its surface [31]. On the other hand, in the case of dry reforming, it was reported that by dispersing $Mo_2C$ on $Al_2O_3$, novel active acidic sites were produced [32]. Nevertheless, according to our knowledge, in the case of HDO, carbides supported over non-carbon materials have received very little attention.

The present work aims to evaluate the catalytic properties of rhenium carbide supported over non-carbon-based materials. Three supports were considered, namely $SiO_2$, $Al_2O_3$, and $ZrO_2$.

## 2. Results and Discussion

### 2.1. Catalysts Characterization

The textural properties obtained from $N_2$-sorption isotherms (Figure 1) are summarized in Table 1. Figure 1 shows that all of the supports presented a type IV isotherm typical of mesoporous materials, although the hysteresis loops were different. A type H2 hysteresis was obtained in the case of $Al_2O_3$ and $ZrO_2$, corresponding to the existence of bottle-neck shape pores, while a type H1 hysteresis was obtained for $SiO_2$, corresponding to the presence of cylindrical shape pores. The pore size distribution obtained from the desorption branch of the isotherm using the BJH model indicates that $ZrO_2$ possessed the smallest pore size (ca. 3 nm), followed by $Al_2O_3$ (ca. 7.5 nm) and then $SiO_2$ (12.5 nm). Importantly, $ZrO_2$ presented a narrow pore size distribution, $SiO_2$ showed a broad pore size distribution, and $Al_2O_3$ revealed a bimodal pore size distribution. Thus, the supports used presented distinct textural properties.

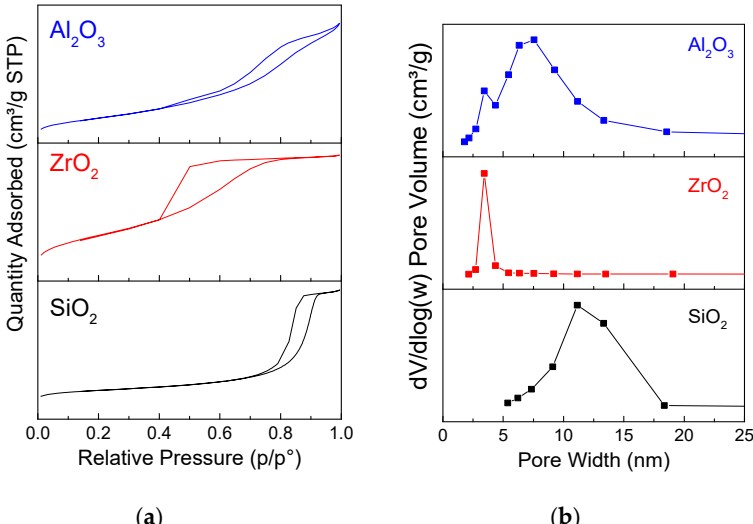

**Figure 1.** $N_2$ adsorption-desorption isotherms (**a**) and pore size distribution (**b**) of the bare supports.

**Table 1.** Textural properties for the supports and the resulting catalysts.

| | | $S_{BET}$ (m² g⁻¹) | Pore Volume (cm³ g⁻¹) | | |
|---|---|---|---|---|---|
| | | | $V_\mu$ | $V_{meso}$ | $V_T$ |
| $Al_2O_3$ | Support | 327 | 0 | 0.43 | 0.43 |
| | Catalyst | 206 | 0 | 0.41 | 0.41 |
| $SiO_2$ | Support | 458 | 0.03 | 1.63 | 1.66 |
| | Catalyst | 278 | 0 | 1.01 | 1.01 |
| $ZrO_2$ | Support | 112 | 0 | 0.09 | 0.09 |
| | Catalyst | 80 | 0 | 0.09 | 0.09 |

Table 1 shows that surface areas of $SiO_2$ and $Al_2O_3$ decreased after the impregnation/carburization of Re. However, the decrease was more pronounced than would be expected for a 10 wt% metal loading, suggesting some degree of pore blockage by Re species. The surface area of $ZrO_2$ also decreased upon Re introduction and the subsequent thermal treatments, although the decrease was less drastic compared to the other catalysts and may be largely due to the presence of Re in the catalyst. This suggests the absence or minimal occurrence of pore blockage and the uniform distribution of Re species in the porous structure of the $ZrO_2$ support.

Figure 2 shows XRD patterns of the catalysts. In the case of ReC/$Al_2O_3$, peaks associated with the support can be observed ($\gamma$-$Al_2O_3$, 00-001-1303), along with weak features around 20° and 30° 2θ, possibly due to rhenium aluminate (00-020-0052). For ReC/$ZrO_2$, peaks corresponding to the tetragonal phase of $ZrO_2$ was identified (00-079-1769), in addition to a diffraction plane related to $Re^0$ (01-087-0599). Notably, there is a shift to higher angle for the $Re^0$ feature (Figure 2b), which has previously been ascribed to incorporating carbon into the matrix during carburization and gives evidence for carbide/oxycarbide formation [28]. In the case of ReC/$SiO_2$, peaks indexed as $Re^0$ phase can be observed. The results suggest that the nature and dispersion of the Re species is influenced by the type of support, with $ZrO_2$ favoring the formation of small carbide/oxycarbide particles, $SiO_2$ favoring the formation of large metallic particles, and $Al_2O_3$ interacting strongly with Re species.

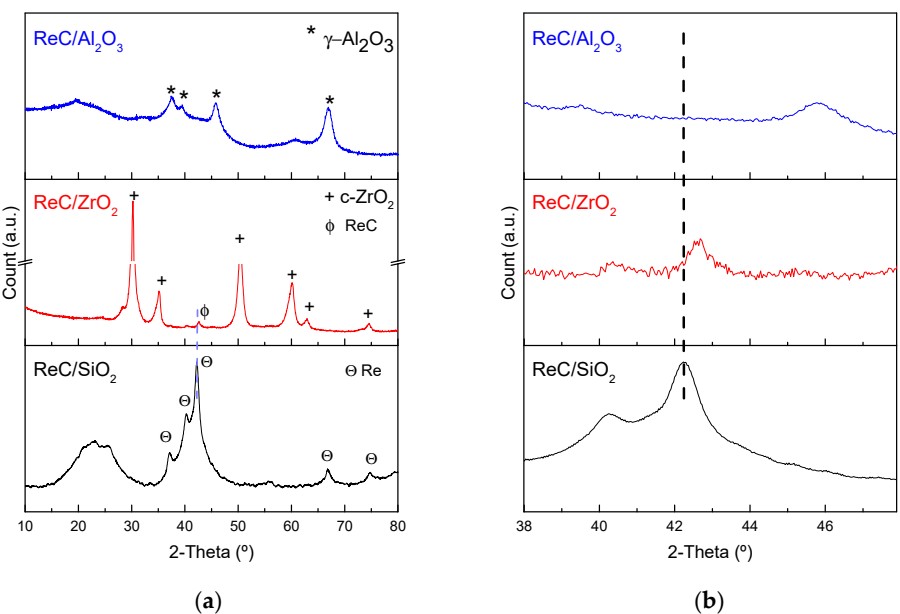

**Figure 2.** Diffractograms obtained for the catalysts (**a**), zoom in the region from 38° to 48° (**b**).

Figure 3 and Table 2 show the surface composition of the catalysts determined by XPS. As noted previously, the passivated catalysts were re-reduced at 350 °C for 30 min under $H_2$ to mimic the conditions during reaction.

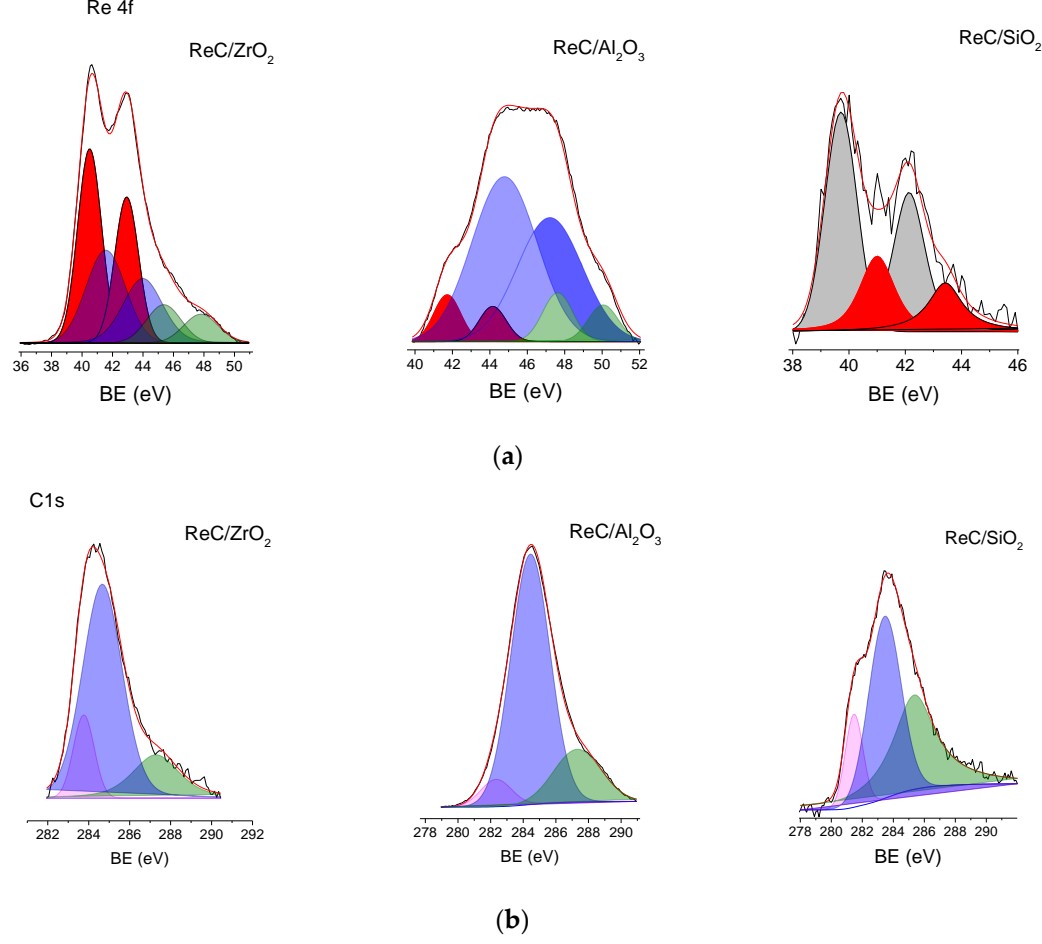

**Figure 3.** XPS deconvolution of the Re 4f (**a**) and C 1s (**b**) component.

**Table 2.** XPS data obtained for the catalysts after in situ re-reduction at 350 °C.

| Catalyst | BE, eV (%) | | | | | | | | |
|---|---|---|---|---|---|---|---|---|---|
| | Re 4f$_{7/2}$ | | | | | C 1s | | | |
| | Re$^0$ | Re$^{\delta+}$ | Re$^{2+}$ | Re$^{6+}$ | Re$^{7+}$ | Carbide | C=C | C-C | C=O |
| **ReC/SiO$_2$** | 40.2 (68) | | 41.5 (38) | | | 282.6 (12) | 284.6 (43) | 286.4 (44) | |
| **ReC/Al$_2$O$_3$** | | | 41.7 (9) | 44.8 (80) | 47.6 (11) | 282.3 (6) | 284.4 (75) | | 287.3 (19) |
| **ReC/ZrO$_2$** | | 40.5 (48) | 41.6 (38) | | 45.4 (14) | 283.7 (13) | 284.6 (68) | | 287.3 (19) |

Table 2 shows that the Re 4f band can be fitted to two components for ReC/SiO$_2$ and to three components for ReC/Al$_2$O$_3$ and ReC/ZrO$_2$. The binding energy (BE) of the different Re species detected were at 40.2, 40.5, 41.6, 44.8, 45.4 and 47.6 ± 0.1 eV. The BE at 40.2 eV can typically be assigned to Re$^0$ species [33], the one at 40.5 eV was previously ascribed to partially carburized Re (Re$^{\delta+}$) [27], the component at 41.6 eV can be ascribed to Re$^{2+}$ species [34], and the BE at 44.8 eV can be attributed to Re$^{6+}$ [34]. The attribution of the remaining species is more complicated as BE of Re is known to shift significantly due to oxygen vacancies [34]. Indeed, the bands at 45.4 and 47.6 eV can both be related to Re$^{7+}$, although the former has been assigned to Re$^{7+}$ in ReO$_4^-$ while the latter has been attributed to Re$^{7+}$ in Re$_2$O$_7$ [33].

The results show clear differences in the nature of the Re species on the surface of the catalysts. The predominant species detected at the surface of ReC/SiO$_2$ was Re$^0$, while the major species on the surface of ReC/ZrO$_2$ was the partially carburized Re, consistent with XRD results. This suggests that the surface and bulk composition of the ReC/SiO$_2$ and ReC/ZrO$_2$ catalysts were similar. The main species present on ReC/Al$_2$O$_3$ was Re$^{6+}$, followed by Re$^{2+}$. Although Re$^{2+}$ and Re$^{6+}$ have been assigned in the literature to ReO and ReO$_3$ species, respectively, they could also be associated with Re$_2$C and ReO$_x$C$_y$ phases, and hence the presence of carbidic phase on ReC/Al$_2$O$_3$ cannot be ruled out.

Figure 3 also shows that the C 1s band can be fitted into three components. The component at 282–283.7 eV can be assigned to carbidic carbon [35], the components at 284.0–284.6 eV and 285.8 eV can be attributed to sp2 (C=C) and sp3 (C-C) carbon, respectively, and finally the component at 287.3 eV can be ascribed to C=O species [36]. The formation of aromatic carbon during carburization is well documented and is as expected [35], and the formation of a carbide phase in the three catalysts can be confirmed. Nevertheless, the BE of the carbidic band appears quite shifted in the case of the ReC/ZrO$_2$ support, which may have been induced by the support. It is known that Zr is the most electropositive of all of the support cations, indicating that it has the highest ability to withdraw electrons from the dispersed species. The result is thus consistent with an electronic modification of the carbide phase in the case of ReC/ZrO$_2$.

Figure 4 displays the TPR profiles of the catalysts. The results consist of the evolution of the TCD signals and the mass fragments corresponding to CH$_4$ ($m/z$ = 15), CO ($m/z$ = 28), and CO$_2$ ($m/z$ = 44) formed during the reduction. The TCD signals correspond to H$_2$ consumed, although it is difficult to obtain clear information on the reduction behavior of carbonaceous species from this result. Here, it is important to remember that the samples are passivated species. Nevertheless, some observations can be made. The TCD signal of the ReC/SiO$_2$ catalyst displays two features at around 150 and 380 °C and a sharp drop at around 600 °C. The feature at 150 °C can be assigned to the reduction of the passivation layer, and the feature at 380 °C can be attributed to the reduction of ReO$_x$ species [11,37]. The sharp drop coincided with the formation of CO and could be associated with the reduction of some oxycarbide species. The TCD signals for ReC/ZrO$_2$ and ReC/Al$_2$O$_3$ also show the same two lower-temperature features observed for the ReC/SiO$_2$ (150 and 350 °C), along with an additional feature around 500 °C. The higher-temperature feature could be due to the existence of particles that interacted more strongly with the support.

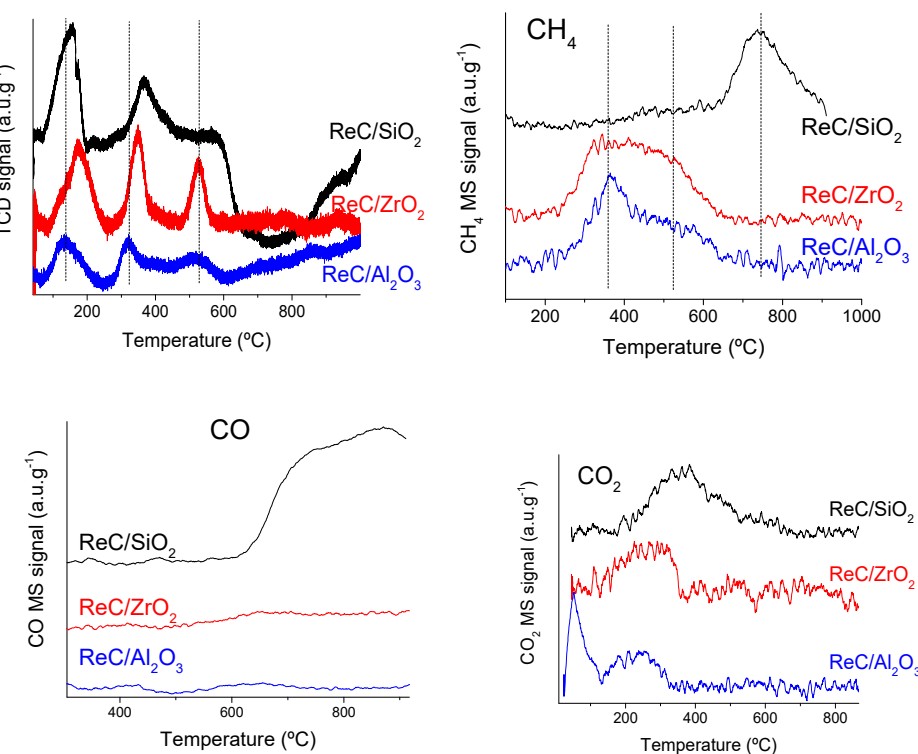

**Figure 4.** Evolution of the TCD signal and MS fragments of $CH_4$, CO, and $CO_2$ formed during TPR.

The $CH_4$ traces give additional insights into the species present on the catalysts. In the case of ReC supported over $Al_2O_3$ and $ZrO_2$, the last feature at around 500 °C is accompanied by $CH_4$ formation. According to the literature, $CH_4$ results from gasification reaction and can be associated with different carbon species depending on the temperature [28,38,39]. For this reason, the $CH_4$ traces were deconvoluted and the results are presented in Figure 5. For the $ReC/Al_2O_3$ and $ReC/ZrO_2$ catalysts, three components could be distinguished: 376, 475, and 552 °C for $ReC/Al_2O_3$; 327, 412, and 520 °C $ReC/ZrO_2$. The first two peaks correspond to carbidic species, whereas the latter is related to pyrolitic carbon [28,38,39]. Thus, two different carbidic phases were detected on the catalysts, although the lower temperatures obtained for $ReC/ZrO_2$ suggest that the particles interacted less strongly with the $ZrO_2$ support than with the $Al_2O_3$ support.

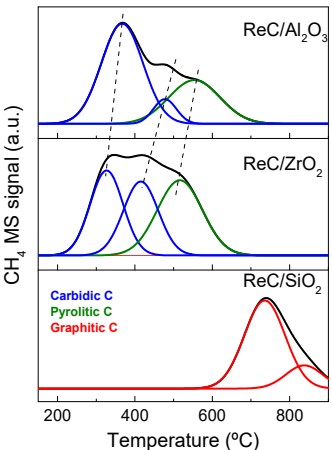

**Figure 5.** Deconvolution of the $CH_4$ MS signal for each catalyst.

Finally, the formation of $CO_2$ could also be appreciated for the three sample with a wide band centered around 400 °C in the case of $SiO_2$, 300 °C for $Al_2O_3$ and in the case of

$ZrO_2$, the band is centered at around 250 °C and present a second band at 100 °C. Those band could be associated to the decomposition of some oxygenated carbon species such as carboxylic or lactonic groups [40] or eventually to the reduction of some oxycarbides species [28].

Figure 6 shows $NH_3$-TPD profiles of the supports and the corresponding catalysts. The results give information on the acidic properties of the catalysts. The desorption profiles for the $SiO_2$ support and the $ReC/SiO_2$ catalysts were featureless, indicating the absence of acidic sites. The profiles for the other supports show broad peaks from 100 to 300 °C, indicating the presence of weak and medium acid sites. [41] After the impregnation and carburization steps, the TPD profile shows that $NH_3$ desorbed between 100 and 200 °C, suggesting that medium acid sites were eliminated likely due to deposition of Re species.

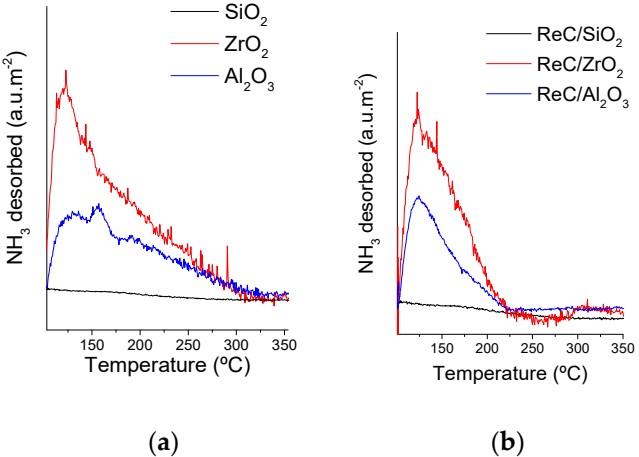

(**a**)                  (**b**)

**Figure 6.** $NH_3$-TPD profile of the supports alone (**a**) and the resulting catalysts (**b**).

Temperature programmed decomposition of isopropylamine (TPD-IPA) was carried out to assess the Brønsted acidity of the supports and the catalyst. It is well-established that IPA decomposes into $NH_3$ and propene over Brønsted acid sites (BAS) through the Hoffmann elimination reaction [42]. Thus, the MS signal of propene ($m/z = 41$) can be used for comparison of the Brønsted acidity and is presented in Figure 7.

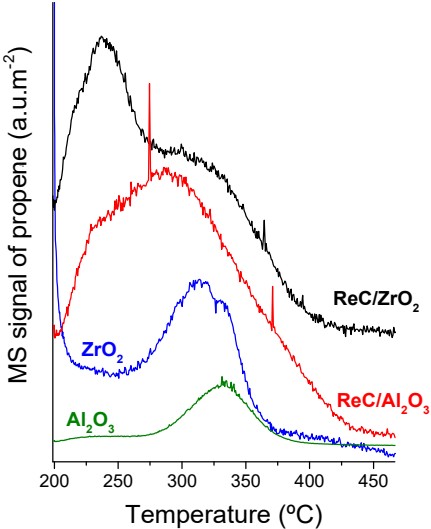

**Figure 7.** TPD-IPA profile of the supports alone and the resulting catalysts.

It can be observed that in the case of the bare supports, propene is formed at around 320 °C for $ZrO_2$ and 334 °C for $Al_2O_3$. The lower temperature observed in the case of

$ZrO_2$ compared to $Al_2O_3$ indicates that the BAS on $ZrO_2$ are stronger than on $Al_2O_3$. After impregnation and carburization of Re, two features of propene are observed at 238 °C and 312 °C in the case of $ZrO_2$ and at 239 °C and 291 °C for $Al_2O_3$. The lower temperature feature is related to strong BAS, whereas the higher temperature feature is due to weak BAS. Interestingly, the addition of the metal increased the amount of BAS on both samples and introduce new acid sites. Additionally, it can be observed that the weaker BAS are more prominent over $ReC/Al_2O_3$, while the stronger BAS are more prominent on $ReC/ZrO_2$.

In summary, catalysts characterization results provide clear evidence of support's influence on the nature and dispersion of the active phase obtained after carburization. Over $SiO_2$, the results suggest that $Re^0$ was mainly obtained and carburization was minor. Over $ZrO_2$, it appears that $Re_xC$ was obtained together with $Re_2C$. Finally, over $Al_2O_3$, the data suggest that $Re_2C$ and $ReC_xO_y/ReO_x$ may have been formed. The origin of these differences could reside in the acidity of the support. The desorption profiles of $NH_3$ indicate that $SiO_2$ has negligible acidity while $Al_2O_3$ and $ZrO_2$ have acid sites of comparable strength. The TPD-IPA results indicate that $ZrO_2$ presented stronger BAS than $Al_2O_3$, which may have played a role in the formation of ReC with a higher amount of strong BAS.

### 2.2. Catalytic Properties

The catalysts and the bare supports were evaluated for guaiacol HDO. Table 3 summarizes the initial rate of conversion ($r^0$) together with the CO uptakes. Negligible guaiacol conversion was obtained over the $SiO_2$ support, while appreciable conversions were observed over the $ZrO_2$ and $Al_2O_3$ supports. The initial rates over the $ZrO_2$ support ($9.5 \times 10^{-6}$ mol g$^{-1}$ s$^{-1}$) was about two times higher than on the $Al_2O_3$ support ($4.8 \times 10^{-6}$ mol g$^{-1}$ s$^{-1}$).

**Table 3.** CO uptake and initial rate of guaiacol conversion obtained at 350 °C.

|  | $SiO_2$ | $ReC/SiO_2$ | $ZrO_2$ | $ReC/ZrO_2$ | $Al_2O_3$ | $ReC/Al_2O_3$ |
|---|---|---|---|---|---|---|
| CO uptake ($\mu$mol g$^{-1}$) | - | 5 | - | 35 | - | 28 |
| $r^0$ ($10^{-6}$ mol g$^{-1}$ s$^{-1}$) | 0 | 1.5 | 9.5 | 14.2 | 4.8 | 9.9 |
| TOF (s$^{-1}$) | - | 0.30 | - | 0.41 | - | 0.35 |

Table 3 shows that the initial rates over the catalysts decreased in the order, $ReC/ZrO_2$ > $ReC/Al_2O_3$ > $ReC/SiO_2$. The turnover frequency (TOF), reported based on sites counted by CO chemisorption, were different, suggesting that the conversion of guaiacol at the initial stage of the reaction was catalyzed by different active sites. Such differences could be expected as the supports are not inert (except $SiO_2$), for instance $ZrO_2$ could account for near 2/3 of initial rate than $ReC/ZrO_2$ while $Al_2O_3$ could account for nearly half of the activity of the $ReC/Al_2O_3$. Interestingly, in the case of $ReC/SiO_2$, a weak CO uptake was measured despite that $Re^0$ was identified as the active phase. One explanation for this could be the presence of graphitic carbon formed during the carburization that might block the access of active site [43–45].

Figure 8 shows a comparison of the product selectivity for the bare supports (Figure 8a) and the catalysts (Figure 8b) at similar conversions. The label ''Other'' includes methylated compounds such as veratrole, methylated guaiacol, etc.

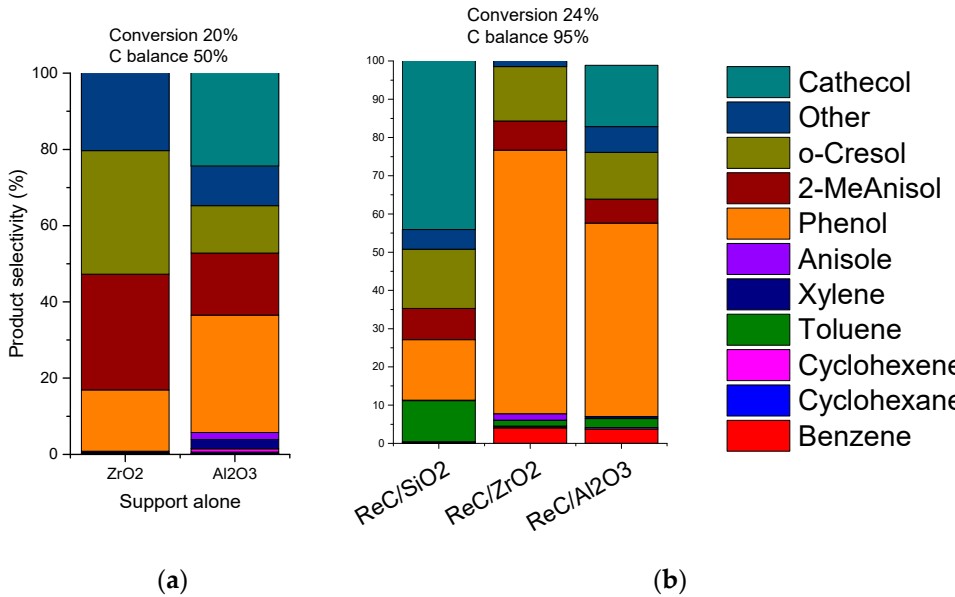

**Figure 8.** Comparison of the product selectivities obtained at 20–24% of guaiacol conversion and the respective carbon balances over the bare supports (**a**) and the catalysts (**b**). Reaction conditions: 200 mg catalyst, 350 °C, 5.0 MPa $H_2$.

Figure 8 shows that a poor carbon balance was obtained over the bare supports, suggesting strong adsorption of guaiacol on the acid sites of the support or rapid formation of coke. The main products formed over $ZrO_2$ were cresol, methylanisole, methylated and methoxy compounds, and phenol. These are generally consistent with reported results in the literature, although cyclohexanone was also reported [46]. These products were also observed over the bare $Al_2O_3$ support. However, catechol was also produced in significant amounts, in addition to anisole, xylene, cyclohexene, which were formed in trace amounts. The results are consistent with those reported in the literature [47,48]. Thus, the results indicate the possible involvement of acid sites on the support in the reaction, although it is not clear whether those sites became inaccessible after the impregnation and carburization of Re species.

The product selectivities over the catalysts show important differences. In the case of $ReC/SiO_2$, the main product obtained was catechol (44%), followed by cresol and phenol (both 15%), toluene (10%), and then methylanisole (8%). The products observed and the selectivities obtained suggest that demethylation and methyl transfer reactions, along with dehydroxylation, occurred significantly. It was previously shown that $Re^0$ was the main surface species present on the $ReC/SiO_2$ catalyst, indicating that metallic Re species are mainly responsible for the reaction. However, the possible participation of $Re^{2+}$ species cannot be disregarded. Direct comparison of these results to previously published of guaiacol HDO over $Re/SiO_2$ is difficult due to the differences in the reaction temperature (300 vs. 350 °C) and the likely existence of carbidic Re species in the present work [33,49].

For the $ReC/ZrO_2$ and $ReC/Al_2O_3$ catalysts, there were some similarities in product selectivities. Both catalysts were most selective towards phenol, the catalyst $ReC/ZrO_2$ (68%) being more selective than $ReC/Al_2O_3$ (50%). In addition, the selectivities to benzene, toluene, 2-methyl-anisole, and o-cresol were similar over the catalysts. The similarities are in line with TPR and in a lesser extent, to XPS results which showed some degree of similarities in the surface species on the catalysts. However, there were some differences, the most notable being the selectivity to catechol over $ReC/Al_2O_3$ and the lack there of over $ReC/ZrO_2$. This difference may have arisen from the direct involvement of acid sites associated with the supports influencing the reaction pathway. It was previously observed (Figure 8a) that while the bare $Al_2O_3$ support was selective to catechol, the bare $ZrO_2$ support was not. Furthermore, the higher selectivity to phenol and the absence of

catechol over the ReC/ZrO$_2$ catalyst suggests that this catalyst possessed sites that allowed bypassing of catechol. It is suggested that the existence of the Re$_x$C phase, as indicated by XPS, played an important role in the direct formation of phenol.

Figure 9 shows the evolution of the products yield and guaiacol conversion. Due to the low conversion, it is difficult to deduce the sequence of products formation from the results for the ReC/SiO$_2$ catalyst. However, it is clear that catechol was formed initially and remained a dominant product during the course of the reaction. This suggests that subsequent conversion of secondary products was minimal (Figure 10). In the case of ReC/ZrO$_2$ and ReC/Al$_2$O$_3$, the yield profiles of phenol and o-cresol are consistent with those of intermediate products. However, it was not clear from the data if they were primary products due to the appreciable guaiacol conversion even at the onset of reaction. What is clear is that catechol was also formed over ReC/Al$_2$O$_3$, and thus it is likely that a fraction of the phenol produced originated from dehydroxylation of catechol.

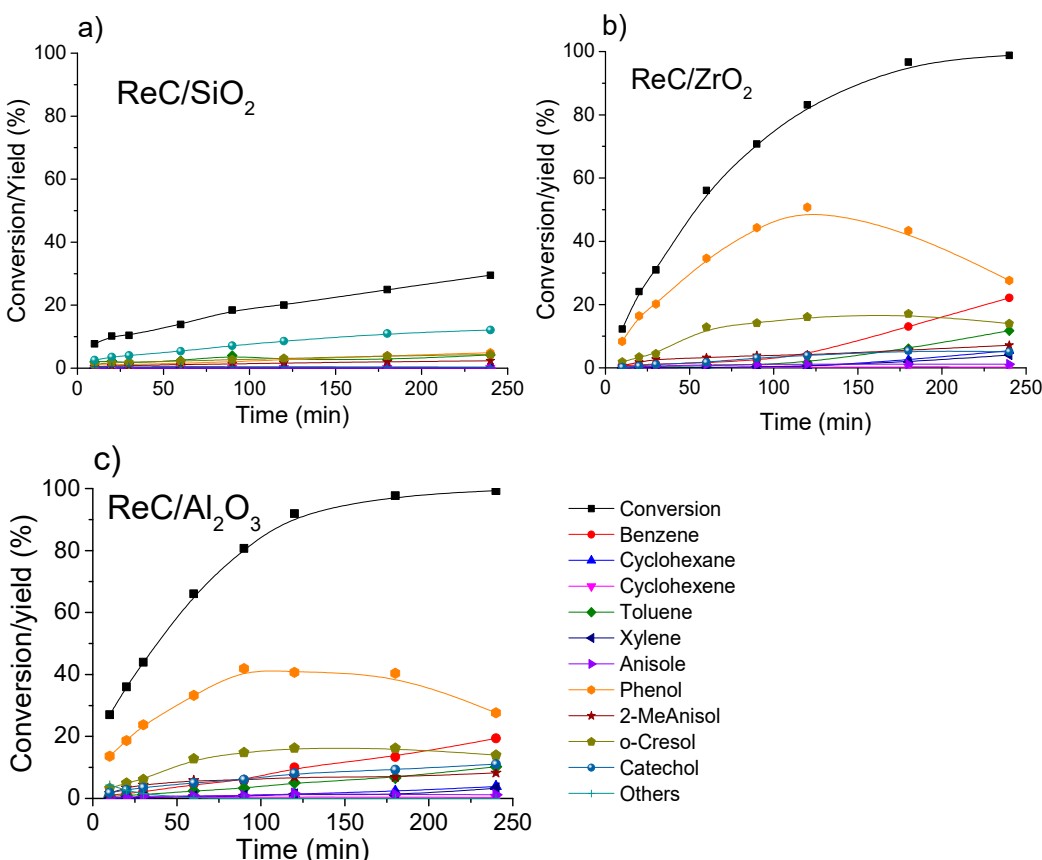

**Figure 9.** Evolution of guaiacol conversion and the products yield for the different catalyst (**a**) ReC/SiO$_2$ (**b**) ReC/ZrO$_2$ and (**c**) ReC/Al$_2$O$_3$. Reaction conditions: 200 mg catalyst, 350 °C, 5.0 MPa H$_2$.

Another important observation is the yield of benzene: it can be observed that benzene was subsequently formed, which is in general agreement with other previous reports concerning carbides [27]. However, there was one important difference between this result and the previous one. In the previous study over ReC/AC, conversion of phenol to benzene only began once guaiacol had been completely converted, which was attributed to the competitive adsorption of phenol with guaiacol [27]. In the present case, conversion of phenol to benzene began when guaiacol conversion reach around 80%. It is reasonable that acid sites present over the ZrO$_2$ and Al$_2$O$_3$ supports may have aided in the adsorption of phenol, facilitating its conversion to benzene even at lower guaiacol conversion. Also, the yield of toluene, likely formed from hydroxylation of o-cresol, increased with increasing

reaction time. It is important to remark that the significant yields of benzene and toluene (aromatic hydrocarbons) and the substantially low yields of cyclohexane and cyclohexene (fully hydrogenated compounds) is indicative of the contribution of the supports in minimizing hydrogenation, particularly when compared to the results of ReC/AC [27].

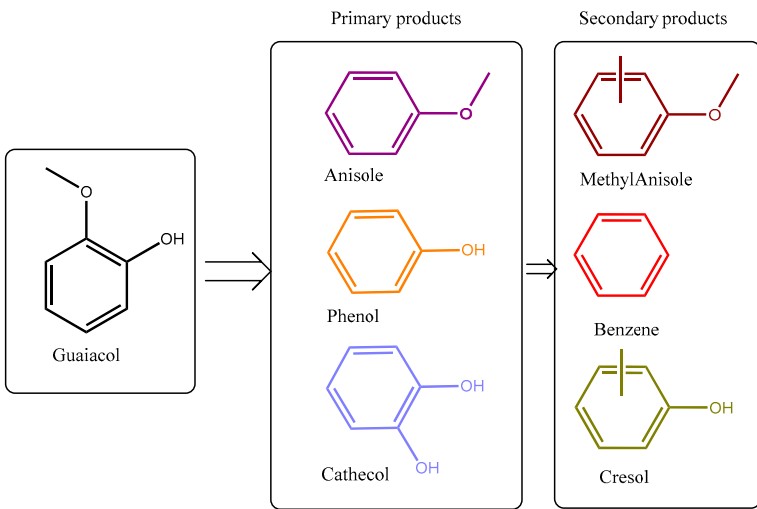

**Figure 10.** Main product obtained from guaicol HDO over Re-based catalysts.

Finally, the evolution of the products that contain 0,1, or 2 oxygen with time are presented in Figure 11 to compare more specifically how efficient are the catalysts in the HDO compared to other reactions. In this figure, others represent methylated products obtained over acid sites. Figure shows that a similar trend is obtained on both supports with some slight differences such as the formation of a lower amount of others over $ZrO_2$ than $Al_2O_3$ at high conversion on the benefit of completely deoxygenated products (ca. 0-oxygen). This figure demonstrated that the carbide obtained over $ZrO_2$ is more selective toward HDO reactions than $Al_2O_3$.

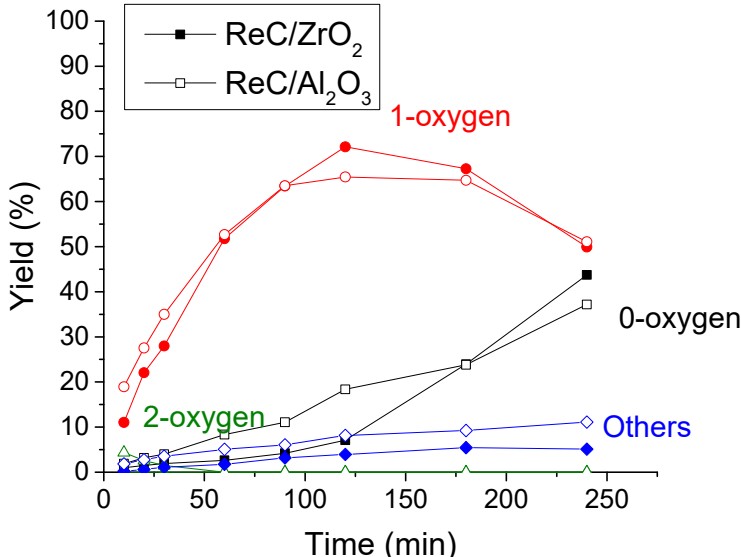

**Figure 11.** Evolution with time of the reactions products groups by their amount of oxygen.

## 3. Materials and Methods

### 3.1. Materials

The chemicals used in this study were $ZrOCl_2$ (99%, Merck, Darmstadt, Germany), $NH_4ReO_4$ (99%, Sigma-Aldrich, Darmstadt, Germany), aqueous $NH_3$ (25%, Loba Chemi

PVT, Mumbai, India), $Al_2O_3$ (SPH 501 A, Rhodia, France), $SiO_2$ (Grace, USA), dodecane, hexadecane, guaiacol (all reagent grade, Merck, Darmstadt, Germany). The gases used were purchased from Indura (Chile).

### 3.2. ZrO$_2$ Synthesis

$ZrO_2$ was prepared by a precipitation method, according to a procedure reported by Znaiguia et al. [50] Typically, $ZrOCl_2$ (0.4 M) was added dropwise to an aqueous solution of $NH_3$ (25%) under stirring and the pH was maintained at 9. The mixture was left for 2 h, filtered, and then washed with deionized water until a neutral solution was obtained. The solid was finally calcined under air at 450 °C for 1 h.

### 3.3. Catalyst Preparation

Catalysts were prepared by incipient wetness impregnation (IWI) of an aqueous solution of $NH_4ReO_4$ over the supports (nominal content of Re was fixed at 10 wt%). The supports were previously ground, sieved ($\leq$150 μm), and dried at 110 °C for 2 h. After impregnation, the mixture was left for 6 h at room temperature for maturation and dried for 15 h at 80 °C. Then, the catalysts were carburized at 650 °C for 1.5 h under a flow (100 mL min$^{-1}$) of 75%$H_2$/25%$C_2H_4$. These conditions were chosen according to the previous studies [27,28]. Finally, the samples were passivated at ambient temperature for 1 h under a flow (50 mL min$^{-1}$) of 5%$O_2$/$N_2$. The catalysts are denoted as ReC/support.

### 3.4. Catalyst Characterization

Textural properties were obtained by $N_2$ sorption isotherms at −196 °C using a 3-Flex device (Micromeritics, Norcross, GA, USA). Prior to the measurements, the samples were degassed at 300 °C for 4 h on a SmartVacPrep equipment (Micromeritics, Norcross, GA, USA). Surface area was obtained from the BET method and pore size distribution was obtained by the BJH model on the desorption branch of the $N_2$ isotherm. The total pore volume ($v_T$) was obtained from the quantity of $N_2$ adsorbed at $p/p^0 = 0.997$ and the micropore volume ($V_\mu$) was determined from the t-plot equation. The mesopore volume ($V_{meso}$) was obtained from the difference between $V_T$ and $V_\mu$.

X-ray diffraction (XRD) measurements were carried out with a Polycrystal X'Pert ProPANalytical instrument (UK) using Ni-filtered Cu Kα radiation λ = 0.15406 nm) and a graphite monochromator. For each sample, Bragg's angles between 4° and 100° were scanned using a step size of 0.04° and a scanning rate of five steps per second.

X-ray photoelectron spectroscopy (XPS) measurements were carried out using an Escalab 200 R spectrometer (Thermo Fisher Scientific, Waltham, MA, USA) equipped with a hemispherical analyzer and a non-monochromatic Mg K X-ray radiation (hν = 1253.6 eV) source. The binding energy (BE) was measured by reference to the C 1s peak at 284.6 eV, with an equipment error of less than 0.01 eV in the energy determinations. The spectra of Re 4f and C 1s were fitted to a combination of Gaussian–Lorentzian curves of variable proportions using the least-square regression. The following constraints were considered for Re 4f fitting: spin-orbit splitting of 2.43 eV, equal FWHM for the 4f 7/2 and 5/2, and a fixed area ratio of 4:3 for those two components.

The same 3Flex device (Micromeritics) was employed for CO chemisorption and temperature programmed reduction/desorption measurements. For the CO chemisorption, the samples were degassed in situ at 110 °C for 30 min and re-reduced at 350 °C for 30 min under $H_2$ flow (60 mL min$^{-1}$). CO chemisorption was carried out at 35 °C and the uptake was determined by subtracting two isotherms recorded consecutively with an evacuation step of 1 h between the two measurements.

Temperature-programmed reduction (TPR), temperature-programmed desorption of $NH_3$ (TPD-$NH_3$) and temperature-programmed decomposition of isoprolylamine (TPD-IPA) were carried out with a combination of a thermal conductivity detector (TCD) and a mass spectrometer (Cirrus 2, MKS Instruments Inc., Andover, MA, USA). For the TPR measurement, the samples were heated at 10 °C min$^{-1}$ under 5% $H_2$/Ar flow

(100 mL min$^{-1}$). A cold trap was placed between the outlet of the reactor and the TCD to remove the water produced during reduction. The effluent stream was monitored by MS and the masses 15 (CH$_4$), 18 (H$_2$O), 28 (CO), and 44 (CO$_2$) were recorded. For the TPD-NH$_3$ measurement, the samples were pretreated under He (100 mL min$^{-1}$) at 500 °C for 30 min. The sample was then exposed to NH$_3$ (30 mL.min$^{-1}$) at 100 °C for 15 min, purged under He (100 mL min$^{-1}$) to remove physisorbed NH$_3$, and then heated under He at a rate of 10 °C min$^{-1}$ while monitoring the $m/z$ = 17 (NH$_3$) signal. For the TPD-IPA measurement, the samples were pretreated under He (100 mL min$^{-1}$) at 500 °C for 30 min, subjected to pulses of IPA at 200 °C, and then heated under He at a rate of 10 °C min$^{-1}$ while monitoring the $m/z$ = 17 (NH$_3$), 41 (propylene) and 44 (IPA) signals.

*3.5. Catalytic Properties*

Guaiacol conversion was carried out in a 100 mL stirred-batch reactor from Parr instrument. The catalyst (0.2 g) was introduced with guaiacol (1.7 g), dodecane (32 g) as the solvent, and hexadecane (700 μL) as the internal standard. The reactor was first purged for 10 min with N$_2$ flow and then heated to 350 °C under autogenous pressure and continuous stirring (645 rpm). An aliquot was first collected, and then hydrogen was introduced to the system to increase the total pressure to 5.0 MPa. Liquid samples were collected periodically during the course of the reaction (4 h) and were analyzed and quantified using a Nexis GC-2030 gas chromatograph (Shimadzu, Kyoto, Japan) equipped with an Elite-1 column (Perkin Elmer, Waltham, MA, USA, 30 m × 0.32 mm, film thickness of 0.25 μm) and an FID detector. The conversion was calculated by two equations (Equations (1) and (2). The first equation is related to guaiacol disappearance, while the second equation is based on product formation. This allowed the carbon balance to be calculated (conversion from Equation (2) conversion from Equation (1)). Product selectivity was calculated by Equation (3), and the initial rate and conversion was obtained by Equation (4). Finally, turnover frequency (TOF) was defined according the sites probe by CO chemisorption (Equation (5)).

$$\text{Conversion (\%)} = \frac{n_{GUA}^0 - n_{GUA}}{n_{GUA}^0} \times 100 \tag{1}$$

$$\text{Conversion (\%)} = \frac{n_{products}}{n_{reagent\ remaining} + n_{products}} \times 100 \tag{2}$$

$$\text{Selectivity (\%)} = \frac{n_{product\ i}}{\sum n_{products}} \times 100 \tag{3}$$

$$r_0 = \frac{b \times n_{GUA}^0}{m} \tag{4}$$

$$\text{TOF} = \frac{r_0}{\text{CO uptake}} \tag{5}$$

In these equations, $n_{GUA}^0$ corresponds to the initial quantity (mol) of guaiacol, $n_{GUA}$ is the quantity at a time t, $n_{products}$ is the sum of all of the products detected (mol), $n_{product\ i}$ is the amount (mol) of a product i, m is the weight (g) of the catalyst, and b corresponds to the initial slope obtained by plotting conversion vs. time (s). In addition, $n_{reagent\ remaining}$ represents the difference between the quantity (mol) of guaiacol at the beginning of the reaction ($n_{GUA}^0$) and at a time t. For the calculation of the initial rates (Equation (4)), only the data at the low conversion region ($\leq$30%) were considered.

**4. Conclusions**

The present work provided strong evidence that the choice of support affects the carburization process. Over an inert support such as SiO$_2$, rhenium was not carburized but instead graphitic carbon was formed and Re$^0$ was the main species detected at the surface. By using acidic supports carburization of rhenium occurred and depending on the surface and chemical identity of the support, different carbidic species were obtained. Over

ZrO$_2$, Re$_x$C was formed, whereas over Al$_2$O$_3$, Re$_2$C was obtained together with oxycarbide species. The most active catalyst was ReC/ZrO$_2$, followed by ReC/Al$_2$O$_3$, and then ReC/SiO$_2$, which was associated to co-participation of sites on the supports. In addition, clear differences in product selectivities were observed over the catalysts, consistent with the existence of different Re species and the possible involvement of the support in dictating the reaction routes. The ReC/SiO$_2$ was most selective to catechol, the ReC/ZrO$_2$ was active for direct formation of phenol, bypassing catechol, and the ReC/Al$_2$O$_3$ was most selective to phenol but also produced catechol. The desirable properties of ReC/ZrO$_2$ were attributed to the existence of Re$_x$C species and Brønsted acid sites on ZrO$_2$. Additionally, all of the catalysts produced an appreciable amount of desirable aromatic hydrocarbons (benzene and toluene), while minimizing excessive hydrogenation of the aromatic ring, an important distinction from previously reported results on activated carbon-supported Re carbides.

**Author Contributions:** Conceptualization, E.B.; methodology, E.B.; validation, E.B., I.T.G., A.B.D. and N.E.; formal analysis, E.B.; investigation, E.B., I.T.G., A.B.D. and N.E.; resources, I.T.G., A.B.D. and N.E.; writing—original draft preparation, E.B.; writing—review and editing, E.B., I.T.G., A.B.D. and N.E.; supervision, N.E.; project administration, E.B.; funding acquisition, E.B. and N.E. All authors have read and agreed to the published version of the manuscript.

**Funding:** This research was funded by Agencia Nacional de Investigación y Desarrollo (ANID) Chile, grants number Fondecyt N° 3170072, N° 1220763, Fondequip N° EQM160070, the Millennium Science Initiative Program—NCN2021_090 and European Union 101008058.

**Data Availability Statement:** Not applicable.

**Conflicts of Interest:** The authors declare no conflict of interest.

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
