# Peer review of "Effect of the Support on Rhenium Carbide in the Hydrodeoxygenation of Guaiacol as Lignin-Derived Model Compound"

_catalysts, doi:10.3390/catal12101229_

Round 1

Reviewer 1 Report

Blanco et al. describe the synthesis of supported Re carbide as catalysts for hydrodeoxygenation of guaiacol in a batch autoclave reactor. When using inert support SiO2 the carbide is not formed, but using acidic supports Al2O3 and ZrO2 the carbide phase is formed and better catalysts are achieved.

The paper is well written and easy to follow. It is an interesting study, which is well performed, and the experimental methods are mostly well described making it possible to reproduce the results. I recommend publication after minor revisions.

Specific comments:

L. 28-29: Give numbers on the different selectivities in the abstract.

Introduction: There are not that many references to the use of carbides as HDO catalysts. E.g. Mortensen, PM, de Carvalho, HWP, Grunwaldt, J-D, Jensen, PA & Jensen, AD 2015, 'Activity and stability of Mo2C/ZrO2 as catalyst for hydrodeoxygenation of mixtures of phenol and 1-octanol', Journal of Catalysis, vol. 328, pp. 208-215. https://doi.org/10.1016/j.jcat.2015.02.002 could be relevant to compare to.

Section 2.5: It is a bit unclear if the autoclave was flushed with H2, after the air purge with N2, in order to eliminate all N2 before pressurizing. Is H2 missing in line 156, where it says “, and then was introduced”? Please clarify.

L. 207: Are the authors sure it is cubic ZrO2. This phase is only stable at very high temperatures. Typical ZrO2 supports are either tetragonal, which is meta-stable, or monoclinic. The XRD of the cubic and tetragonal phases for high surface area materials (small crystalize sizes) are very similar, except distinctions in peak intensities. Consider if the ZrO2 used is in fact tetragonal.

For the catalytic activity, besides the specific selectivities to intermediate and final products as function of time, it would be good to show the degree of deoxygenation (fraction of organic bound oxygen converted to water, CO or CO2) as function of time. This will likely make it more clear that the ZrO2 supported catalyst is superior to the Al2O3 supported catalyst.

The background colors in Fig.9 makes it unclear.

A major drawback of using carbides as HDO catalysts is that they could oxidize and deactivate in the presence of water. A short discussion of this and how this may be investigated in a future study could be relevant.

Reviewer 2 Report

Questions were listed in attachment. Please check. Authors should also polish language and check gramma of the manuscript.

Round 2

Reviewer 2 Report

Authors provided reasonable solution of problems I listed. Authors also added more information in their manuscript. The future work is expected to understand more insightfully of the catalysis. Although there are some typos in sentences and minor spelling errors, this manuscript can be accepted.